# Hospital inpatient care utilization among patients with tuberculosis, Republic of Ireland, 2015–2018

James O'Connell[1]*, Eoghan de Barra[1,2], Samuel McConkey[1,2]

1 Department of International Health and Tropical Medicine, Royal College of Surgeons in Ireland, Dublin, Republic of Ireland, 2 Health Service Executive, Beaumont University Hospital, Dublin, Republic of Ireland

* jamesoconnell@rcsi.com

**Data Availability Statement:** All relevant data are within the manuscript and its Supporting Information files.

## Abstract

### Background

The Republic of Ireland (ROI) has a low incidence of TB. A reform of TB services in 2003 recommended that the delivery of care to patients with TB should primarily be in the outpatient setting, with limited indications for hospitalization. Three hospitals were designated as TB centres. Our aim was to describe the utilization of hospital inpatient care by patients with TB in the ROI.

### Methods

We searched public hospital coding data to identify discharges between 01/01/2015-31/12/18 where TB was the principal diagnosis. The cost of TB episodes of care was calculated using payment rules for public hospitals in the ROI.

### Results

We identified 1185 discharges with TB as the principal diagnosis. Of these, 68% (801/1185) were emergency episodes of care and 32% (384/1185) were elective. We estimate that 65.1% (818/1257) patients with TB notified in the ROI from 2015 to 2018 who had an episode of care in a public hospital was 65.1% (818/1257) and that 50.8% (639/1257) of those notified had an emergency episode of care. The estimated mean annual cost of TB inpatient care per year in the ROI from 2015 to 2018 was €2,638,828–2,955,047, with emergency episodes of care having a mean annual cost of €2,250,926–2,557,397 per year.

### Conclusions

The burden of TB on hospital inpatient care in the Republic of Ireland is significant.

**Funding:** This study was funded by the Royal College of Surgeons in Ireland. The funders had no role in study design, data collection and analysis, decision to publish, or preparation of the manuscript. The salary of Dr James O'Connell was funded by the Royal College of Surgeons in Ireland.

**Competing interests:** The authors have declared that no competing interests exist.

## 1. Introduction

The Republic of Ireland (ROI), like many countries, has a long and difficult history controlling tuberculosis (TB). In the first half of the 19[th] century, the rapid growth in the population, the poor social conditions, and the effects of the Famine of 1845–1849 resulted in a national epidemic of tuberculosis [1,2]. This epidemic persisted for an entire century. By 1945, the mortality rate from TB was 125 cases per 100,000 population. In 1948, the State embarked upon the largest program of public hospital construction in its history [3]. The result was the creation of several sanitoria and over 2,000 additional hospital beds for TB care [3]. The establishment of a mass radiography screening program aided the identification and isolation of TB cases. This coupled with the advent of anti-tuberculosis medication and a national BCG vaccination program brought the epidemic under control by the end of the 1950s [3]. The incidence of TB has since declined from 230 cases per 100,00 population in 1953 to 6.6 cases per 100,000 population in 2018 [4].

The World Health Organisation (WHO) End TB Strategy has set the target of a 90% reduction in global TB incidence by 2035 [5]. It is estimated the ROI will need an annual reduction of 19% to meet this target, far greater than the current 3.1% annual reduction [6]. The WHO outlined the challenges for meeting this goal in low incidence countries including effective TB service planning and delivery [6]. A challenge for health systems is maintaining TB services and clinical expertise as the incidence of TB declines to low levels [7].

In the ROI there is no national TB strategy as exists in other low-incidence countries, nor is there a defined budget for TB care [8]. There are national clinical guidelines [9]. The health system in the ROI compromises of both private and public healthcare. While 43.3% of the population in Ireland have private health insurance, only 14% of all healthcare expenditure is funded from private health insurance [10,11]. In the public health service, outpatient consultations are provided at no charge to patients. Funding for TB medications are provided by the public health service, except for a €2.00 prescription charge [12]. Primary care is provided by 2,954 general practitioners (GP) [13]. An appointment with a GP is free for patients entitled to it based on a means assessment [12]. There is a fee to attend the emergency department without referral by a GP [14]. In this system, patients with symptoms of TB can present to their general practitioner or the emergency department. Patients presenting to their general practitioner with symptoms of TB can, if deemed necessary by the GP, be referred to the emergency department or an appropriate outpatient clinic depending on their clinical presentation. If a diagnosis of TB is established by a physician, they are legally obliged to notify it as a case to their local director of public health [15]. They can also, but are not obliged to, refer the patient to a physician in an outpatient service who has expertise in the management of TB cases. The role of the public health departments in the ROI comprises maintaining TB surveillance and performing contact tracing of notified cases where necessary. They also oversee directly observed therapy for TB patients. There are many routes through which a patient can be diagnosed with TB in the ROI which will include one or a combination of general practitioners, public health doctors and nurses, hospital outpatient and inpatient services, emergency departments and the private healthcare system.

The last reform of TB services in Ireland was outlined in a 2003 report [16]. It was recommended that most TB care should be delivered on an outpatient basis in acute general hospitals with a small number of beds allocated in three hospitals for inpatient TB management. This resulted in three hospitals being designated as TB centres. Consultant staffing with a special interest in TB in all three hospitals was recommended. One of these, St. James's hospital, was designated a supra-regional TB centre with co-location of the national tuberculosis reference laboratory. These hospitals were to be allocated resources to provide adequate recreation and

rehabilitation facilities to serve patients with TB, including those in other non-TB centres. The national TB guidelines in Ireland state that a minority of tuberculosis patients should require hospital admission [9]. The guidelines also advise treatment of TB should be directed by a respiratory/infectious diseases physician with training in the management of TB [9].

Achieving WHO End TB targets and eliminating TB from the ROI will require purposeful action by the state in the same vein as bringing the epidemic under control did. Evaluating healthcare resource utilisation by patients admitted to hospital with TB may provide insight into how services could be improved. We aimed to describe the utilization of hospital inpatient care by patients with TB in the ROI.

## 2. Methods

### 2.1 Data sources

All patients who have an episode of care in a public hospital in the ROI have their diagnostic data collected and coded according to the International Classification of Diseases-10-Australian Modification (ICD-10-AM) $8^{th}$ edition using a computerised system known as the Hospital Inpatient Enquiry (HIPE) system [17]. It is the principal source of demographic, administrative and clinical data on all episodes of care in publicly funded hospitals in the ROI. An episode of care is defined as beginning when the patient is admitted to the hospital and ends when the patient is discharged or dies [18].

The National Quality Assurance and Improvement System (NQAIS) Clinical is an online interactive application that analyses HIPE data to provide detailed information to clinicians and managers [19]. Episodes of care can be searched based on the principal diagnosis. The principal diagnosis is defined as the diagnosis established to be chiefly responsible for occasioning the episode of care [18]. Episodes of care can be further described based on the length of stay and whether the episode was emergency or elective. Elective episodes of care are any scheduled hospital admission [18]. This includes scheduled attendances to the hospital for procedures such as bronchoscopies, and also scheduled admissions for inpatient hospital care. Emergency episodes of care are any unscheduled admission to a public hospital [18]. This includes unscheduled admissions to the hospital through an emergency department, medical assessment unit or unscheduled transfers between hospitals. Patients attending the outpatient department or the emergency department who do not get admitted to hospital are not captured by the HIPE system.

Comparative data on national TB case notifications were extracted from national TB surveillance reports for 2015–2018 [4,20–22].

### 2.2 Data extraction

The NQAIS Clinical was searched for all episodes of care where TB was the principal diagnosis and where discharge occurred between 01/01/2015 to 31/12/18. We extracted for every episode of care the patient identification number, age, sex, county of origin, postcode, admitting hospital, admission source, discharge destination, admission type (emergency or elective), principal speciality, length of stay, length of stay in intensive or coronary care, secondary diagnoses, readmission information and Charlson Comorbidity Index, The Charlson Comorbidity index is a method of quantifying morbidity and predicting mortality by classifying or weighting between 0 and 17 comorbid conditions [23]. Patients with an index of 0 are said to have none of the comorbidities contained in the Charlson Comorbidity Index, while for example, patients with moderate chronic kidney disease will have an index of 2.

## 2.3 Data analysis

Across public hospitals in the ROI there is no unique patient identification number. This means the same patient can have one or multiple identification numbers within NQAIS Clinical if they have attended more than one hospital. We performed matching of episodes of care based on age, gender, county of residence, postal code, admission source, discharge destination and the patient identification numbers extracted from NQAIS Clinical to identify the number of patients in our cohort.

We searched data extracted to identify cases of respiratory TB and non-respiratory TB using the ICD-10 AM codes shown in Table 1 [24]. We also searched secondary diagnoses and place of residence for codes relating to homelessness, illicit drug use, imprisonment, diabetes mellitus, alcohol use or recent TB case contact. The European Centre for Disease Control (ECDC) advise that these variables, known as social determinants and risk factors (SDRFs) for TB, should be identified and monitored because they are thought to be important factors in the development of TB disease among people in Europe [25]. We searched the patients' records for secondary diagnoses which may inform potential reasons for hospital admission. These included sepsis, systemic inflammatory response syndrome, common side effects of antimycobacterial treatments, non-adherence to medication, isolation, and drug resistance.

The cost of all TB inpatient episodes of care was estimated using costing guidance from the Healthcare Pricing Office [26]. We extracted the diagnosis-related grouping (DRG) for every episode of care coded on NQAIS Clinical as having TB as the principal diagnosis. DRGs are a means of classifying patient hospital encounters into a manageable number of groups which

**Table 1. Terms used to search NQAIS clinical.**

| Diagnosis | ICD-10 Code/Term Used |
|---|---|
| Tuberculosis (all) | A15.0–15.9, A16.0–16.9, A17.0, A17.1, A17.8, A17.9, A18-18.8, A19.0, A19.1, A19.2, A19.8, A19.9, B90.0, B90.1, B90.2, B90.8, B90.9, J65, M01.10–19, M49.0, M49.00–49.09, N33.0, N74.0, N74.1, P37.0, Z06.74, Z86.11 |
| Respiratory tuberculosis | A15.0–15.9, A16.0–16.9 |
| Diabetes | E10.0–10.9, E11.0–119, E13.0–13.9, E14.0–14.9 |
| Homelessness | Z59.0, admission source |
| Illicit drug use | Z72.2, F11.0–11.9, F14.0–14.9 |
| Alcohol misuse | F10.0–10.9, K70.0-K70.9 |
| Being a prisoner | Prison stated as admission source or discharge destination |
| HIV | B20-24, Z21, R75, O98.7 |
| Contact with and exposure to tuberculosis | Z20.1 |
| Sepsis or end-organ failure or volume depletion | A41.0–41.9, R57.2, R65.0–0.3, R65.9 |
| | J96.0, J96.1, J96.9, K72.0, K72.9, N17.9, F05.9, E86, I50.1, R41.0 |
| | Admission to intensive or coronary care coded |
| Adverse effect of antimycobacterial drug | Y41.1, Y40.6 |
| Rash, skin eruption | R21, L27 |
| Nausea and vomiting, gastroenteritis and colitis | R11, A09.9, K52.1,52.3, 52.8.52.9 |
| Abnormal liver function tests | R94.5 |
| Visual disturbance, optic neuritis | H53.0–3.9, H46 |
| Isolation | Z29.0 |
| Resistance to antimycobacterial drug | Z06.6, Z06.7 |
| Non-adherence to medication | Z91.1 |

can be used to describe the mix of cases being managed by a hospital [26]. The DRGs group together cases which are clinically similar, and which are expected to consume a similar amount or resources. Each DRG can be further subdivided based on the level of complexity consumed during the admission into high, intermediate, and low complexity. Complexity refers to the intensity of the resource utilization rather than the clinical complexity. For example, a patient with a terminal illness being managed palliatively may be considered clinically complex but may have a low level of resource utilization and be referred to as low complexity. The cost of an episode of care was calculated using the DRGs, the level of complexity, the length of stay and the type of episode of care. Where the DRG complexity classification was missing from the data extracted from NQAIS Clinical we calculated the cost assuming first, the episode of care was of high complexity and then assuming it was of low complexity. This provides an upper and lower limit cost value for that episode of care.

Data analysis was performed using Microsoft Excel and Stata 16.0 (StataCorp. 2015). Using each patient's first emergency episode of care we performed logistic regression to compare differences in length of stay and cost between subgroups (patients with respiratory and non-respiratory TB, patients with and without SDRFs for TB, patients with a Charlson Comorbidity Index of zero and a Charlon Comorbidity Index of greater than zero). A chi-square test was performed to examine the relationship between categorical variables such as gender, disease site and the presence of SDRFs for TB. Ethical approval was received from the Royal College of Surgeons in Ireland Research Ethics Committee to perform this analysis of HIPE data and costing of episodes of care (reference number 1633).

## 3. Results

### 3.1 Description of episodes of care

Between 2015 and 2018 there were 1185 episodes of care with TB as the principal diagnosis, 76.9% (911/1185) of which were in patients with respiratory TB (Table 2). Emergency episodes of care made up 67.6% (801/1185) of all episodes of care, of which 76% (609/801) were in patients with respiratory TB. From 2015 to 2018 there were 1257 TB cases notified in the ROI. There were 818 patients identified after matching of episodes of care. We estimate that the proportion of TB cases notified who had an episode of care was 65.1% (818/1257). Of these, 73.1% (615/841) of cases of respiratory TB notified had an episode of care compared to 48.8% (203/416) of cases of non-respiratory TB. We estimate that 50.8% (639/1257) of cases notified had an emergency episode of care. Of these, 57.3% (482/841) of cases of respiratory TB notified had an emergency episode of care compared to 37.7% (157/416) of non-respiratory TB. Most of these patients had only one emergency episode of care (respiratory TB, 80.7% (389/482), non-respiratory TB 83.4% (131/157)).

60.7% (674/1108) of all episodes of care in adults aged 16 years and over occurred outside of the three hospitals deemed to be TB centres of care (Table 3). 53% (589/1108) of episodes of care in patients aged 16 years and over were under specialists other than TB specialists.

741/801 (80%) of emergency episodes of care had a Charlson Comorbidity Index of 0. 131/801 (16.4%) of emergency episodes of care had a side effect potentially attributable to anti-mycobacterial usage coded. 110/801 (13.7%) had a requirement for isolation of any type coded. 146/801 (18.2%) of emergency episodes of care had sepsis, systemic inflammatory response syndrome, volume depletion, end-organ dysfunction or a requirement for intensive care/coronary care coded. 21/801 (2.5%) had non-adherence to a medication coded. 18/801 (2.2%) had resistance to an antimycobacterial coded. 503/801 (61.5%) of episodes of care had none of these coded.

**Table 2. Description of episodes of care.**

| Episodes of Care on NQAIS Clinical, 2015–2018 | | | |
|---|---|---|---|
| | All | Respiratory TB | Non-respiratory TB |
| Episodes of Care (proportion of all episodes of care) | 1185 | 911 (76.9%) | 274 (23.1%) |
| Emergency (proportion of all episodes of care) | 801 | 609 (76.1%) | 192 (23.9%) |
| Elective (proportion of all episodes of care) | 385 | 302 (78.4%) | 83 (21.6%) |
| Tuberculosis Cases Notified, Republic of Ireland, 2015–2018 | | | |
| | All | Respiratory TB | Non-respiratory TB |
| Cases (proportion of all cases notified) | 1257 | 841 (66.9%) | 416 (33.1%) |
| Patients on NQAIS Clinical, 2015–2018 | | | |
| | All | Respiratory TB | Non-respiratory TB |
| Patients (proportion of all cases notified) | 818 (65.2%) | 615 (73.1%) | 203 (48.8%) |
| Patients who had an emergency episode of care (proportion of cases notified) | 639 (50.8%) | 482 (57.3%) | 157 (37.7%) |
| Patients who had an elective episode of care (proportion of cases notified) | 287 (23%) | 220 (26.2%) | 67 (16.1%) |

## 3.2 Patient characteristics

61% (499/818) of patients were male (Table 4). This is similar to the proportion of males among TB cases notified (59.2% (744/1257)). The median age of patients was 43 years (interquartile range 23–60) and the age distribution of patients is similar to that of all TB cases notified over the same period (Fig 1). 75.2% (615/818) of patients had respiratory TB and 24.8% (203/818) of patients had non-respiratory TB. The proportion of patients with respiratory TB in our cohort is higher than that of the proportion of TB case notified (66.9% (841/1257)). The median ages of patients with respiratory and non-respiratory TB were equivalent (43 years (IQR 29–60) vs 43 years (IQR 29–61)). Males made up 62% (381/615) and 58.1% (118/203) of respiratory and non-respiratory TB patients, respectively.

16.9% (138/818) of patients had a SDRF for TB. The median age of patients with SDRF for TB was greater than that of patients without SDRF for TB (51 years (43–63) vs 40 years (IQR 28–59), P < .001, S1 Table in S1 File). Males made up a higher proportion of patients with SDRF for TB compared those without SDRFs for TB (79% (109/138) vs. 57.4% (390/680)). The relation between gender and having SDRFs for TB was significant, ($\chi^2$ (1, N = 138) = 22.57, P < .001, S1 Table in S1 File). A higher proportion of respiratory TB patients had a SDRF for TB compared to non-respiratory TB patients (18.7% (115/615) vs 11.3% (23/203)) and this relation was significant ($\chi^2$ (1, N = 818) = 5.9098, P = .02, Table 4). This was mainly due to a higher proportion of alcohol misuse in those with respiratory TB (8% (49/615) vs 2% (4/203), P = .002, Table 4). The most prevalent SDRFs for TB were diabetes, 7.3% (60/818), and alcohol misuse, 6.5% (53/818).

**Table 3. Episodes of care by specialist and centre.**

| | Episodes of Care | Emergency episodes of care (proportion of emergency episodes of care) | Elective episodes of care (proportion of elective episodes of care) |
|---|---|---|---|
| Age 16 years and over | 1111 | 763 | 348 |
| TB Centre (in those age 16 years and over) | 437 (39.3%) | 258 (33.8%) | 179 (51.4%) |
| Non-TB centre (in those age 16 years and over) | 674 (60.7%) | 505 (66.2%) | 169 (48.6%) |
| Principal speciality-TB specialist (in those aged 16 years and over) | 522 (47%) | 333 (43.6%) | 189 (54.3%) |
| Principal speciality-Non-TB specialist (in those aged 16 years and over) | 589 (53%) | 430 (56.4%) | 159 (45.7%) |

**Table 4. Characteristics of patients with TB from NQAIS clinical and TB cases notified, Republic of Ireland, 2015–2018.**

| Tuberculosis Cases Notified, Republic of Ireland, 2015–2018 | | | | |
|---|---|---|---|---|
| | All | Respiratory TB | Non-respiratory TB | Comparison |
| Total (proportion of all cases notified) | 1257 | 841 (66.9%) | 416 (33.1%) | |
| Male sex (proportion of all cases notified) | 744 (59.2%) | N/a | N/a | |
| Drug resistant TB | 66/1257 (5.3%) | N/a | N/a | |
| Patients on NQAIS Clinical, 2015–2018 | | | | |
| | All | Respiratory TB | Non-respiratory TB | |
| Total (proportion of all patients) | 818 | 615 (75.2%) | 203 (24.8%) | |
| Median age (interquartile range) | 43 (23–60) | 43 (29–60) | 43 (29–61) | |
| Male sex (proportion of total) | 499 (61%) | 381 (62%) | 118 (58.1%) | |
| Drug resistant TB | 14/818 (1.7%) | 13 (2.1%) | 1 (<1%) | |
| HIV (proportion of total) | 28 (3.4%) | 21 (3.4%) | 7 (3.4%) | |
| Any SDRF for TB (proportion of total) | 138 (16.9%) | 115 (18.7%) | 23 (11.3%) | Respiratory TB versus non-respiratory TB P = 0.02 |
| Diabetes (proportion of total) | 60 (7.3%) | 44 (7.2%)) | 16 (7.9%) | |
| Alcohol misuse (proportion of total) | 53 (6.5%) | 49 (8%) | 4 (2%) | Respiratory TB versus non-respiratory TB p = 0.002 |
| Illicit drug use (proportion of total) | 18 (2.2%) | 17 (2.3%) | 1 (<1%) | |
| Incarceration (proportion of total) | 3 (<1%) | 2 (<1%) | 1 (<1%) | |
| Contact with and exposure to TB (proportion of total) | 10 (1.2%) | 10 (1.6%) | 0 | |
| Homelessness (proportion of total) | 15 (2.2%) | 14 (2.3%) | 1 (<1%) | |
| Died during an episode of care (proportion of total) | 18 (1.7%) | 13 (2.1%) | 5 (2.4%) | |

N/a = Not available.

## 3.3 Cost of episodes of care with a principal diagnosis of TB

In total 16,005 bed-days were utilised across 38 hospitals by patients with a principal diagnosis of TB, of which 87.9% (14060.5/16005) were during an emergency episode of care. The cost of

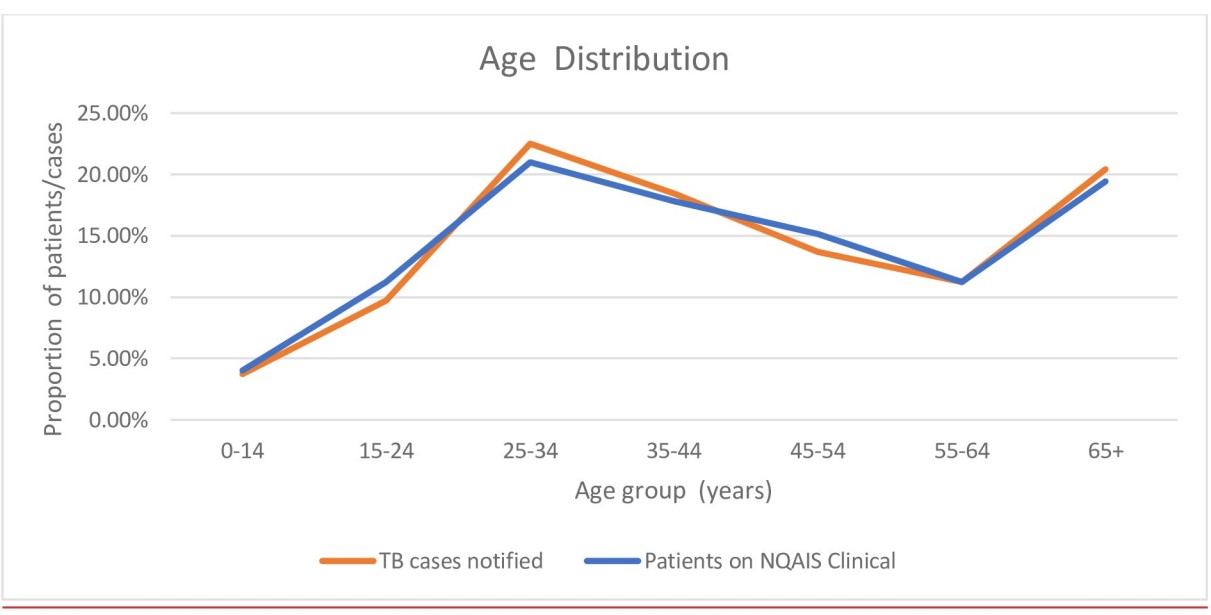

**Fig 1. Age distribution of TB patients on NQAIS clinical and TB cases notified nationally.**

**Table 5. Cost of episodes of care in respiratory and non-respiratory TB patients.**

| | Respiratory TB | | | Non-respiratory TB | | |
|---|---|---|---|---|---|---|
| | Emergency episodes of care | Elective episodes of care | All episodes of care | Emergency episodes of care | Elective episodes of care | All episodes of care |
| Total (proportion of all episodes of care) | 609 (67.1%) | 299 (32.9%) | 908 | 192 (69.3%) | 85 (30.7%) | 277 |
| Number of patients | 482 | 220 | 615 | 157 | 67 | 203 |
| Number of bed-days 2015–2018 | 10387 | 1382.5 | 11769.5 | 3673.5 | 562 | 4235.5 |
| Median length of stay (days) (IQR) | 9 (5–19) | 0.5 (0.5–2) | 6 (0.5–15) | 12.5 (6–21) | 0.5 (0.5–7) | 8 (2–18) |
| | Cost (lower-upper limit) | Cost (lower-upper limit) | Cost (lower-upper limit) | Cost (lower-upper limit) | Cost (lower-upper limit) | Cost (lower-upper limit) |
| Total cost (€) | 5,927,708–7,090,711 | 1,091,440–1,131,536 | 7,019,148–8,222,247 | 3,075,934–3,138,876 | 459,062–460,169 | 3,536,103–3,597,938 |
| Mean cost per year (€) | 1,481,927–1,772,678 | 272,860–282,884 | 1,754,787–2,055,562 | 768,984–784,719 | 114,765–115,042 | 884,023–899,485 |
| Mean cost per episode (€) | 9,734–11,643 | 3,650–3,784 | 1,933–2,264 | 16,021–16,348 | 5,401–5,414 | 3,191–3,247 |
| Mean cost per case notified (€) | 7,048–8,431 | 1,298–1,345 | 8,346–9,777 | 7,394–7,545 | 1,104–1,106 | 8,500–8,649 |
| Mean cost per bed-day (€) | 571–683 | 789–818 | 596–699 | 837–854 | 817–819 | 835–849 |

all episodes of care was €10,555,312–11,820,186, of which 85.3–86.5% (€9,003,702–10,229,587) was attributable to emergency episodes of care. The estimated average cost of TB hospital care per year in the ROI from 2015 to 2018 was €2,638,828–2,955,047, with emergency episodes of care costing an average of €2,250,926–2,557,397 per year.

11769.5/16005 (73.5%) of bed-days were utilized by patients with respiratory TB. Patients with a higher cost of emergency episodes of care per thousand euro were less likely to have respiratory TB than non-respiratory TB (odds ratio (OR) 0.978, 95% CI 0.966–0.991, P = .001, S3 Table in S1 File). Patients with non-respiratory TB had a higher cost per bed-day than patients with respiratory TB (€837–854 vs. €571–683, Table 5). There was no association between the length of stay and whether the patient had respiratory TB or non-respiratory TB (OR 0.994, 95% CI 0.987–1.001, P = .111, S3 Table in S1 File).

Patients with a longer length of stay for emergency episodes of care were more likely to have SDRFs for TB (OR 1.017, 95% CI 1.009–1.025, P<0.001, S4 Table in S1 File) and were more likely to have a Charlson Comorbidity Index greater than zero (OR 1.024, 95% CI 1.015–1.033, P < .001, S5 Table in S1 File). Patients with a higher cost of emergency episodes of care per thousand euro were also more likely to have SDRFs for TB (OR 1.01, 95% CI 1.001–1.019, P = -.034, S4 Table in S1 File) and were more likely to have a Charlson Comorbidity Index of greater than zero (OR 1.045 95% CI 1.028–1.062, P < .001 S5 Table in S1 File).

When comparing patients with drug resistant and drug sensitive TB there was no difference in the cost of emergency episodes of care per thousand euro (OR 1.009, 95% CI .994–1.023, P = .233, S6 Table in S1 File) but the length of stay was greater in those with drug resistant TB (OR 1.016, 95% CI 1.004–1.027, P = 0.004, S6 Table in S1 File).

## 4. Discussion

Our study shows there is a high need for inpatient care among patients with TB in the ROI. A higher proportion of patients with respiratory TB appear to require unscheduled emergency inpatient care than patients with non-respiratory TB. Patients with non-respiratory TB had

more costly emergency episodes of care. We demonstrated that patients with longer and more costly emergency episodes of care were more likely to have SDRFs for TB and to have comorbidity (as measured by the Charlson Comorbidity Index). Whether this is related to more clinically complex episodes of care or the need for enhanced support to enable timely discharge is unclear from our data. We found that most episodes of care occurred outside of TB centres and not under the direct care of a TB specialist. This pattern of care is at variance with our national guidelines and policy [9,16]. The annual cost of emergency episodes of care in patients with TB was high (€2.25m-2.57m) considering the incidence of TB in the ROI is low. From our cohort, we found most emergency episodes of care did not have a common antimycobacterial side effect, non-adherence or drug resistance coded as a secondary diagnosis. This suggests that emergency episodes of care were not related to these factors in most cases. The proportion of patients coded as having sepsis, end-organ dysfunction or a need for intensive/coronary care was also low, suggesting patients presenting with critical illness was not a significant factor either.

A dependence on hospital inpatient services in the management of TB patients and the associated high costs is not unique to the ROI. TB services in other low-incidence countries have described similar experiences. In Germany, a reduction in the overall cost of TB disease to the state was attributed to a reduction in the proportion of TB patients requiring inpatient care from 80% to 71.2% over 5 years [27]. A 2010 study in New York City found that 72% of their cohort required hospital admission, many of which could have been avoided [28]. A national survey in Italy found that 71.6% of all TB cases were hospitalised [29].

It is difficult to ascertain what value is derived from such a high and costly utility of unscheduled emergency care given that treatment outcomes are unknown for a large proportion of TB cases in the ROI [30]. In 1959 a significant study in TB care demonstrated that patients with TB could be treated at home with similar outcomes to treatment in hospitals [31]. Many tuberculosis programs have since aimed to provide care without the need for prolonged hospitalisation [28,29]. The high cost of inpatient care in the ROI and the potential for TB disease transmission in institutional settings should act as an impetus to minimizing the hospitalization of patients with TB unnecessarily. Research examining the needs of TB patients seeking emergency care should be performed and outpatient services should be optimized to meet these needs where possible.

The WHO guidance on Directly Observed Treatment (DOTs) states that a key component of a DOTs program is a standardized recording and reporting system that allows assessment of treatment results for each patient and of the TB control programme overall [32]. We propose establishing a regular national cohort review of all TB cases in the ROI to ensure there is a process of quality improvement, improved reporting of TB outcomes and specialist involvement in care provided to all patients. Cohort review has been beneficial to case management elsewhere, such as in the United Kingdom [33]. In parallel with this, we propose a national TB leader, with regional coordinators supported by a national multi-disciplinary team (MDT), including public health specialists, community and primary care doctors and nurses, those treating latent TB, diagnostic laboratory staff, scientists working on tuberculosis control and the service managers. A national TB lead and MDT would follow the WHO DOTs guidance, which states that governments should demonstrate a commitment to sustained TB control activities [32]. While the last policy on TB services in Ireland recommended three tertiary referral hospitals as TB centres, our study shows that TB patients present to many hospitals and are managed under many specialities. This suggests that a model of TB governance in Ireland not geographically restricted to three sites is needed.

There may be a need to provide dedicated funding to outpatient TB services in the ROI. Public hospitals in the ROI are funded based on activities performed as opposed to treatment

outcomes [26]. Healthcare funding models based on activities performed have resulted in an over-dependence on inpatient TB care in other settings [34]. Addressing funding mechanisms for TB care is important in addressing this. Linking the funding received by TB centres for out-patient services to treatment outcomes for TB patients in their region may have a role in reducing the number of episodes of care in hospitals. This would likely be less expensive for the health service [27,34].

A strength of this study is that it captures data on TB admissions from all acute public hospitals in the ROI across 4 years. This study had several limitations. An audit of coding standards in the ROI found that HIPE data lacked specificity compared to international standards, which may lead to under-reporting of clinical complexity [35]. The primary reason for this was difficulty in extracting information from poorly structured medical records. Therefore, there may be an under-representation of TB complexity and cost within the HIPE dataset. This may be improved on with the national roll-out of an electronic health record making information easier to interpret and code [36]. There is no unique patient identifier across all public hospitals in the ROI. We performed a rigorous matching process based on age, gender, county of residence and postal code. There is a possibility this process matched episodes of care by two different patients to one patient, resulting in an underestimation of the number of patients. There is an ongoing process to implement a unique patient identifier across all hospital in the ROI which when fully implemented would overcome this limitation in any future evaluations [37]. This study was limited by the absence of data on TB patients managed in private hospitals in the ROI, which are not captured by the HIPE system. It does not capture presentations to hospital emergency departments which do not result in hospital admission. The patient's country of birth is also not captured which is relevant when discussing TB in low-incidence countries. A significant limitation is that it was not possible to crossmatch cases notified nationally with admissions coded in HIPE data, meaning we could only provide estimates of the proportion requiring admission.

## 5. Conclusion

There is a significant burden on the acute hospital inpatient service from tuberculosis in the Republic of Ireland. The national TB policy should change in recognition of this.

## Supporting information

**S1 File.**
(DOCX)

## Acknowledgments

I would like to acknowledge the support of Healthatlas Ireland, who operate the NQAIS Clinical, in conducting this study.

## Author Contributions

**Conceptualization:** James O'Connell, Eoghan de Barra, Samuel McConkey.

**Data curation:** James O'Connell.

**Formal analysis:** James O'Connell.

**Methodology:** James O'Connell, Eoghan de Barra.

**Supervision:** Eoghan de Barra, Samuel McConkey.

**Writing – original draft:** James O'Connell.

**Writing – review & editing:** Eoghan de Barra, Samuel McConkey.

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
