## [Decision Letter · Decision Letter 0]

5 May 2020

PONE-D-20-10438

The Utilization of Hospital Inpatient Care due to Tuberculosis, Ireland, 2015-2018

PLOS ONE

Dear Dr O Connell,

Thank you for submitting your manuscript to PLOS ONE. After careful consideration, we feel that it has merit but does not fully meet PLOS ONE’s publication criteria as it currently stands. Therefore, we invite you to submit a revised version of the manuscript that addresses the points raised during the review process.

We would appreciate receiving your revised manuscript by Jun 19 2020 11:59PM. To enhance the reproducibility of your results, we recommend that if applicable you deposit your laboratory protocols in protocols.io, where a protocol can be assigned its own identifier (DOI) such that it can be cited independently in the future. For instructions see: http://journals.plos.org/plosone/s/submission-guidelines#loc-laboratory-protocols

We look forward to receiving your revised manuscript.

Kind regards,

Wen-Jun Tu

Academic Editor

PLOS ONE

Journal Requirements:

1. Please amend either the abstract on the online submission form (via Edit Submission) or the abstract in the manuscript so that they are identical.

Reviewers' comments:

Reviewer's Responses to Questions

**Comments to the Author**

1. Is the manuscript technically sound, and do the data support the conclusions?

Reviewer #1: Yes

Reviewer #2: Partly

2. Has the statistical analysis been performed appropriately and rigorously? 

Reviewer #1: Yes

Reviewer #2: Yes

3. Have the authors made all data underlying the findings in their manuscript fully available?

Reviewer #1: No

Reviewer #2: Yes

4. Is the manuscript presented in an intelligible fashion and written in standard English?

Reviewer #1: Yes

Reviewer #2: Yes

5. Review Comments to the Author

Reviewer #1: The authors have described the utilization of hospital inpatient care by TB patients in the Republic of Ireland to understand how well reforms to TB services introduced in 2003 have been applied and whether new reforms are indicated. The study is a worthy undertaking and has identified the need for several important reforms were Ireland to meet its TB elimination targets in a cost-effective manner (the two actions summarized in the discussion are judged to be critical; the second dependent upon implementation of the first if program costs are to be minimized). Several questions, suggested revisions are made below.

Major:

1. Some explanatory text or background information that puts the TB program of Ireland into context is necessary. It is assumed that patients are diagnosed in the community and then sent to the hospital for admission/connection to TB care. Would these be "unscheduled" admissions? Most hospital admissions were emergency admissions and emergency rooms are notoriously bad at diagnosing TB. What is the role of the public health department - who oversees the program? How are patients that are not sent to the hospital processed? Who is responsible for post-hospital care? How are in-hospital and out-of-hospital care connected. Most emergency admissions had a principal diagnosis of respiratory TB (74.%) - from a public health - infection control/occupational health and safety - perspective it would be very important for the emergency room/hospital to know in advance whether these patients were coming. Were proper precautions taken/did all the model facilities have respiratory isolation capacity - did this influence the designation of three TB centres in 2003?

2. The results section could be a little clearer - a lot of information is presented and it relates to individuals and admissions; with one individual having the potential for multiple admissions. it would be helpful if the authors created at least one additional TABLE and two or three FIGUREs. The TABLE might contain information on the age, sex, country of birth (Ireland vs Foreign-born) and disease site (respiratory and non-respiratory) of (i) their notified cases and (ii) the unique cases in their admissions dataset. The FIGUREs might contain a breakout of their 735 emergency admissions according to the results provided in sections 3.1-3.3. Most of the text refers to the 735 emergency admissions; but in places (line 162 and the FIGURE) the authors include the 67 patients who were same-day admissions/discharges.

3. The authors need to clarify how they managed the admission if TB itself was the secondary diagnosis.

4. Did the authors have access to information about the HIV/AIDS status of their patients and the drug-resistance patterns in the isolates from those that were culture positive; these variables would almost certainly have an influence on length of stay, model of hospital and the type of physician caring for the patient.

Minor:

1. The Charlson score should be defined for readers who may not be familiar with it.

Reviewer #2: This is an interesting manuscript on a topic of local importance. However, I have some serious concerns about data that are missing from this analysis, without which the conclusions and policy recommendations will be deeply flawed.

My main concerns are with the following 2 points: First, the authors do not separate out or address in their discussion the fact that they found significant increases in hospitalizations and duration of hospitalization in people with social determinants of TB, which include homelessness, incarceration, alcohol use disorders, among others. In many places, these patients end up admitted and staying longer because of these factors. This may be very important for the costing considerations--would someone homeless or incarcerated be able to be discharged "home" as rapidly as those who are not? Patients with alcohol use disorders are also often kept longer or followed more closely to ensure that they do not have dangerous side effects. And often, these social determinants keep people from presenting with symptoms earlier, before the situation becomes an emergency. Second, a huge limitation for me with this paper is that there is no differentiation made between people with drug sensitive and drug-resistant disease. I don;t know what the policies are in Ireland, but in many countries, MDR patients require hospitalization for several weeks. Without knowing the reality of these situations, policy recommendations (such as "Reductions in the number of hospital

admissions of TB patients in Ireland should be pursued" line 236) may be misguided.

Minor concerns:

Lines 110-113: the authors seem to define emergency admissions twice--once with same day stay and once with overnight stay. Could it be clarified that these are different endpoints and will both be considered?

Line 121: please define charlson score for non-clinician readers, & give some info about how it's used. What's a possible range of scores? What does a score of 0 imply or mean vs. a score of 20?

Lines 118-131--the authors talk about their analysis methods but it's not clear what they are comparing with the chi2 tests.

Line 136 definition of "complexity" is quite vague. What does it mean to have resource complexity? An example here would help.

Line 142: Is this ethics review for the whole study, or just the costing piece? If whole study, put this in a separate paragraph. Also, the ID number of the Ethics committee would be great to include if you have it.

Lines 158 to 160: are the number of bed-days distributed Gaussian or are the data skewed? If skewed, which I suspect is sort of usual for TB stay durations, I'd suggest presenting the median and IQR instead of the mean. Or present both. mean bed-days as a descriptive measure could be deceptive if you have a few very long stays and most quite short.

6. PLOS authors have the option to publish the peer review history of their article (what does this mean?). If published, this will include your full peer review and any attached files.

Reviewer #1: No

Reviewer #2: No

---

## [Author Response · Author response to Decision Letter 0]

27 May 2020

Dear Editor and Reviewers,

Thank you for reviewing our manuscript and providing constructive comments. We present a considerably improved version of our original manuscript which hopefully you find acceptable. Our response to reviewers’ comments are outlined below.

Have the authors made all data underlying the findings in their manuscript fully available?

Reviewer #1: No

Reviewer #2: Yes 

We have uploaded a Microsoft Excel file which contains the data underlying our results. We have included additional tables which describe the data underlying our results. We hope this demonstrates that the results are based on technically sound analysis and interpretation of the data. While the original manuscript costed only respiratory TB episodes of care we now include the costing of all TB episodes of care using the same methodology. The data underlying the costing is available in the supplementary excel file.

Reviewer 1

Some explanatory text or background information that puts the TB program of Ireland into context is necessary

We have included a detailed description of the health system in the Republic of Ireland in the introduction section and also explained how the TB service is arranged in this context.

It is assumed that patients are diagnosed in the community and then sent to the hospital for admission/connection to TB care. 

We agree that ideally the patient would be diagnosed in primary care and referred to the TB clinic for management however there is little data collected nationally to support or refute this assumption. An unpublished clinical audit from our own tertiary referral centre found over an 18 month period 16/40 patient diagnosed with TB first presented with their symptoms directly to the Emergency Department, 18/40 were attending their primary care physician of which 5/18 were sent by their primary care physician to the emergency department. These figures, in our region at least, would refute the assumption that the majority of TB is diagnosed in primary care. It is not possible to ascertain the reason for an episode of care from the national coding data outside of what the principal diagnosis was.

Would these be "unscheduled" admissions? 

We have elaborated in the methods section what is mean by our use of the term “unscheduled”. If the patient attends the Emergency Department, either with or without a referral by their primary care doctor then this is “unscheduled”. Scheduled or planned hospital admissions such as those for procedures e.g. a bronchoscopy are referred to as “elective”. We have standardised our usage of terminology throughout the manuscript using the terms “emergency”, “elective” and “episode of care” and defined these terms in the methods section. 

Most hospital admissions were emergency admissions and emergency rooms are notoriously bad at diagnosing TB.

Patients who have an episode of care coded are those who attended the emergency department and were admitted to hospital. They are seen by an emergency department doctor and referred to the internal hospital doctor for hospital admission if it is agreed it is necessary. The data recorded and reported on in this study does not describe patients who were seen by an emergency department doctor and not referred for admission. This as you have rightly pointed out that patients with undiagnosed TB could present to an emergency department and not have their diagnosis established. As such, data reported in this study more than likely is an under representation of patients with TB seeking care by attending a hospital emergency department.

What is the role of the public health department - who oversees the program?

We have included a description of the role of public health departments in TB care. They largely are not involved in the assessment of people with signs and symptoms of TB outside of contact tracing of known cases. While in other countries TB clinics can have strong involvement of public health departments, in the Republic of Ireland clinics are typically operated by infectious diseases and respiratory physicians in tertiary care centres. We have commented on this in the introduction section.

How are patients that are not sent to the hospital processed?

Typically, patients will be referred to an outpatient clinic operated by a specialist with expertise in TB, usually but not necessarily one of the 3 TB services operating in the country. We have commented on this in the introduction section 

Who is responsible for post-hospital care? How are in-hospital and out-of-hospital care connected.

Post-hospital care is the responsibility of the primary are doctor unless the patient has been referred to an outpatient service on discharge from the hospital. We have tried to describe the health system and TB service in Ireland throughout the introduction and methods section using the questions posted by reviewer 1 as guidance in the introduction section. 

Most emergency admissions had a principal diagnosis of respiratory TB (74.%) - from a public health - infection control/occupational health and safety - perspective it would be very important for the emergency room/hospital to know in advance whether these patients were coming. Were proper precautions taken/did all the model facilities have respiratory isolation capacity - did this influence the designation of three TB centres in 2003?

All hospitals seeing acute medical patients in Ireland would have at least 1 negative pressure isolation room. Whether this was available or utilized appropriately for patients admitted with TB can not be deduced from our hospital coding data. We searched for a requirement for isolation in the secondary diagnoses and found it was not coded often, which could suggest it was either not required or it was not used appropriately or there is under reporting of its use within the coding data. The designation of the three TB centres in 2003 was based on the regional incidence of TB, the geographic location of the hospitals, their role as tertiary referral centres within the regions and the existing expertise in TB care within these hospitals. 

The results section could be a little clearer - a lot of information is presented and it relates to individuals and admissions; with one individual having the potential for multiple admissions.

We have heavily refined our results section. It now includes a costing of all TB episodes of care. We present first a description of the episodes of care found from our search and then a description of the patient characteristics. Overall we think it is clearer to the reader.

it would be helpful if the authors created at least one additional TABLE and two or three FIGUREs. The TABLE might contain information on the age, sex, country of birth (Ireland vs Foreign-born) and disease site (respiratory and non-respiratory) of (i) their notified cases and (ii) the unique cases in their admissions dataset. 

We have included one table which describes the patient cohort as suggested. Unfortunately, data on country of birth is not available from our hospital coding data so this important comparison could not be made.

The FIGUREs might contain a breakout of their 735 emergency admissions according to the results provided in sections 3.1-3.3. Most of the text refers to the 735 emergency admissions; but in places (line 162 and the FIGURE) the authors include the 67 patients who were same-day admissions/discharges.

In our original manuscript we made a distinction between 735 emergency admissions which had a length of stay >/= 1 day and 67 admissions which were discharged on the same day as admission. For simplicity we have removed this distinction from the manuscript by referring to all of these admissions together as “emergency episodes of care”. We have included a table which shows the breakdown of admissions.

The authors need to clarify how they managed the admission if TB itself was the secondary diagnosis.

The aim of our study was to determine the utilization of hospital care due to tuberculosis. In our coding data the principal diagnosis is the diagnosis which resulted in the patient requiring hospitalization. Had we searched the secondary diagnoses of all patients we would have found additional episodes of care in patients who currently have TB or had TB at an unknown point in the past. These admissions would not have been due to TB and whether the diagnosis of TB is current or historical would not have been possible to determine. For these reasons searching the principal diagnoses provides the most accurate cohort of patients to represent hospital care utilization due to TB. 

Did the authors have access to information about the HIV/AIDS status of their patients and the drug-resistance patterns in the isolates from those that were culture positive; these variables would almost certainly have an influence on length of stay, model of hospital and the type of physician caring for the patient.

Yes, we have included the proportion of patients with HIV and drug resistance coded, both were quite low. It may be the case that these results were not available at the time of discharge from hospital when coding of the patients record took place. We demonstrated that patients with drug resistance had more costly episodes of care.

The Charlson score should be defined for readers who may not be familiar with it.

We have defined the Charlson Score for readers.

Reviewer #2: This is an interesting manuscript on a topic of local importance. However, I have some serious concerns about data that are missing from this analysis, without which the conclusions and policy recommendations will be deeply flawed.

My main concerns are with the following 2 points: First, the authors do not separate out or address in their discussion the fact that they found significant increases in hospitalizations and duration of hospitalization in people with social determinants of TB, which include homelessness, incarceration, alcohol use disorders, among others. In many places, these patients end up admitted and staying longer because of these factors. This may be very important for the costing considerations--would someone homeless or incarcerated be able to be discharged "home" as rapidly as those who are not? Patients with alcohol use disorders are also often kept longer or followed more closely to ensure that they do not have dangerous side effects. And often, these social determinants keep people from presenting with symptoms earlier, before the situation becomes an emergency. 

We have acknowledged the higher cost and longer length of stay in patients with social determinants and risk factors for TB. We agree the issue may be one or a combination of delayed presentation, delayed discharge due to a lack of a suitable destination and more clinically complex inpatient episodes of care. It was not possible for us to determine from coded data which of these hypotheses explains our finding. We have acknowledged this in the discussion

Second, a huge limitation for me with this paper is that there is no differentiation made between people with drug sensitive and drug-resistant disease. I don;t know what the policies are in Ireland, but in many countries, MDR patients require hospitalization for several weeks. Without knowing the reality of these situations, policy recommendations (such as "Reductions in the number of hospital

admissions of TB patients in Ireland should be pursued" line 236) may be misguided.

We found the proportion of patients with drug resistance coded was low in our cohort (1.7% of all patients compared to 5.3% of cases notified). We agree that patients with typically require MDR-TB prolonged inpatient care. From 2015- 2018 in Ireland there were 17 cases of MDR-TB notified. The vast majority of patients have pan sensitive (94.7%) or monoresistant TB (3.9%), as such we have not focused our policy interventions on MDR-TB. It is also likely they are under-represented within the coding data. This may be due to the unavailability of TB culture results at the time of discharge from hospital.

Lines 110-113: the authors seem to define emergency admissions twice--once with same day stay and once with overnight stay. Could it be clarified that these are different endpoints and will both be considered?

In our original manuscript we made a distinction between 735 emergency admissions which had a length of stay >/= 1 day and 67 admissions which were discharged on the same day as admission. For simplicity we have removed this distinction from the manuscript by referring to all of these admissions together as “emergency episodes of care”.

Line 121: please define charlson score for non-clinician readers, & give some info about how it's used. What's a possible range of scores? What does a score of 0 imply or mean vs. a score of 20?

We have defined the Charlson Score for readers and provided an example.

Lines 118-131--the authors talk about their analysis methods but it's not clear what they are comparing with the chi2 tests.

We have elaborated on our description of our statistical analysis to describe what we associations and comparison we are making.

Line 136 definition of "complexity" is quite vague. What does it mean to have resource complexity? An example here would help.

We have elaborated that complexity refers to the resources consumed during an episode of care as opposed to the clinical complexity of a case. We have included an example

Line 142: Is this ethics review for the whole study, or just the costing piece? If whole study, put this in a separate paragraph. Also, the ID number of the Ethics committee would be great to include if you have it.

The ethics review approved all components of the study. I cannot find an ID number for the ethics committee, I have inserted the reference number for my ethics application to that committee.

Lines 158 to 160: are the number of bed-days distributed Gaussian or are the data skewed? If skewed, which I suspect is sort of usual for TB stay durations, I'd suggest presenting the median and IQR instead of the mean. Or present both. mean bed-days as a descriptive measure could be deceptive if you have a few very long stays and most quite short.

The data is skewed. We agree and we have now presented the median and interquartile range. 

I hope I have addressed your concerns above and look forward to hearing your decision.

---

## [Decision Letter · Decision Letter 1]

16 Jun 2020

PONE-D-20-10438R1

The Utilization of Hospital Inpatient Care due to Tuberculosis, Republic of Ireland, 2015-2018

PLOS ONE

Dear Dr. O Connell,

Thank you for submitting your manuscript to PLOS ONE. After careful consideration, we feel that it has merit but does not fully meet PLOS ONE’s publication criteria as it currently stands. Therefore, we invite you to submit a revised version of the manuscript that addresses the points raised during the review process.

We look forward to receiving your revised manuscript.

Kind regards,

Wen-Jun Tu

Academic Editor

PLOS ONE

Reviewers' comments:

Reviewer's Responses to Questions

**Comments to the Author**

1. If the authors have adequately addressed your comments raised in a previous round of review and you feel that this manuscript is now acceptable for publication, you may indicate that here to bypass the “Comments to the Author” section, enter your conflict of interest statement in the “Confidential to Editor” section, and submit your "Accept" recommendation.

Reviewer #1: All comments have been addressed

Reviewer #2: All comments have been addressed

2. Is the manuscript technically sound, and do the data support the conclusions?

Reviewer #1: Yes

Reviewer #2: Yes

3. Has the statistical analysis been performed appropriately and rigorously? 

Reviewer #1: Yes

Reviewer #2: Yes

4. Have the authors made all data underlying the findings in their manuscript fully available?

Reviewer #1: Yes

Reviewer #2: Yes

5. Is the manuscript presented in an intelligible fashion and written in standard English?

Reviewer #1: (No Response)

Reviewer #2: Yes

6. Review Comments to the Author

Reviewer #1: There were unfortunate limitations to the datasets used by the authors. It is suggested that, in the Discussion, they mention this limitation and that efforts be made to address it, so that in the future they can track, yet more completely, the effect of their reforms.

Reviewer #2: I appreciate the opportunity to re-review this manuscript and appreciate the authors' attention to and responses to my earlier comments. I believe the revisions to be complete. I have a few additional comments about the revision.

1) although in the text, the authors present evidence of statistical significance through the use of p-values, these are not presented in the tables at all. I strongly suggest that they be presented in a column in the tables; standard practice, as far as I know.

2) One of the assumptions of the Mann-Whitney U test is independence of observations. If you have 800 or so people but 1180 episodes of care, at least some of those are going to be on the same individuals, and thus may violate this assumption. You may want to include a sentence in your limitations section about that.

3)line 229: unbalanced number of parentheses

4)Table 2, respiratory TB/Elective box unbalanced parentheses

5) line 263 and beyond--ensure you are using a "chi" symbol, and not an "X"

6) Table 6, no SDRFs columns are missing some percentages

7. PLOS authors have the option to publish the peer review history of their article (what does this mean?). If published, this will include your full peer review and any attached files.

Reviewer #1: No

Reviewer #2: No

---

## [Author Response · Author response to Decision Letter 1]

2 Aug 2020

Date Typed 03/08/2020

Dear Editor and Reviewers,

Thank you for once more taking the time to review our submission and providing rigorous feedback. Our responses to the comments are outlined below. We hope the changes submitted are sufficient to meet the standard of publication.

1. “There were unfortunate limitations to the datasets used by the authors. It is suggested that, in the Discussion, they mention this limitation and that efforts be made to address it, so that in the future they can track, yet more completely, the effect of their reforms.”

Yes, we agree this is the prime limitation in the research method. We have included a comment which highlights this limitation. We feel that our matching process based on age, gender, county of residence and postal code could potentially match episodes of care to the same patient when in fact they may have been due to different patients. This means the number of patients in the dataset is, if anything, an underestimation meaning the problem highlighted of a high proportion of cases notified requiring hospitalization may even be higher than estimated in this paper. We highlight an already ongoing project in the Republic of Ireland to provide a unique patient identifier across all public hospitals which would overcome this problem when evaluating the service in the future.

The second notable limitation is that the hospital coding data collection process in the ROI is imperfect and lacks specificity i.e. when the coder was uncertain of how to code an episode of care they allocate it to a non-specific ICD-10 code and they may allocate a diagnosis related grouping of lower complexity and therefore lower cost. We cite a national audit which found the primary reason for these discrepancies is that coders were unable to find the relevant information in poorly maintained hospital paper records. We highlight an ongoing national project to rollout an electronic health record to all public hospitals. 

2. “although in the text, the authors present evidence of statistical significance through the use of p-values, these are not presented in the tables at all. I strongly suggest that they be presented in a column in the tables; standard practice, as far as I know”

We have included a column labelled “comparison” which includes the comparison made and the associated p-value. In the supplementary file a column for P values is also included in each table.

3. “One of the assumptions of the Mann-Whitney U test is independence of observations. If you have 800 or so people but 1180 episodes of care, at least some of those are going to be on the same individuals, and thus may violate this assumption. You may want to include a sentence in your limitations section about that”

We agree with this statement. We have re-analysed our data. We have performed logistics regression instead of a Mann-Whitney U test to compare length of stay and cost of episodes of care between patients with respiratory and non-respiratory TB, patients with SDRFs for TB and those without SDRFs for TB and those with comorbidity and those without comorbidity (as measured by the Charlson Comorbidity Index). We have performed this analysis for all patients first episode of care to ensure there is independence in the observations analysed. We have included the results of these analyses, including p-values, in the supplementary file. 

4.“line 229: unbalanced number of parentheses”

We have corrected this.

5.“Table 2, respiratory TB/Elective box unbalanced parentheses”

We have corrected this. 

6.“line 263 and beyond--ensure you are using a "chi" symbol, and not an "X"”

We have corrected this.

7.“Table 6, no SDRFs columns are missing some percentages”

We have corrected this.

---

## [Editor Report · Decision Letter 2]

11 Aug 2020

Hospital Inpatient Care Utilization Among Patients with Tuberculosis, Republic of Ireland, 2015-2018

PONE-D-20-10438R2

Dear Dr. O Connell,

We’re pleased to inform you that your manuscript has been judged scientifically suitable for publication and will be formally accepted for publication once it meets all outstanding technical requirements.

Kind regards,

Wen-Jun Tu

Academic Editor

PLOS ONE
---

## [Editor Report · Acceptance letter]

18 Aug 2020

PONE-D-20-10438R2 

Hospital Inpatient Care Utilization Among Patients with Tuberculosis, Republic of Ireland, 2015-2018 

Dear Dr. O'Connell:

I'm pleased to inform you that your manuscript has been deemed suitable for publication in PLOS ONE. Congratulations! Your manuscript is now with our production department. 

Kind regards, 

on behalf of

Dr. Wen-Jun Tu 

Academic Editor

PLOS ONE